# Population differences in vaccine responses (POPVAC): scientific rationale and cross-cutting analyses for three linked, randomised controlled trials assessing the role, reversibility and mediators of immunomodulation by chronic infections in the tropics

Gyaviira Nkurunungi ,[1] Ludoviko Zirimenya,[1] Agnes Natukunda,[1] Jacent Nassuuna,[1] Gloria Oduru,[1] Caroline Ninsiima,[1] Christopher Zziwa,[1] Florence Akello,[1] Robert Kizindo,[1] Mirriam Akello,[1] Pontiano Kaleebu,[1] Anne Wajja,[1] Henry Luzze,[2] Stephen Cose,[1,3] Emily Webb,[4] Alison M Elliott,[1,3] POPVAC trial team

GN, LZ and AN contributed equally.

► http://dx.doi.org/10.1136/bmjopen-2020-040426

**Correspondence to**
Dr Gyaviira Nkurunungi;
Gyaviira.Nkurunungi@mrcuganda.org

## ABSTRACT

**Introduction** Vaccine-specific immune responses vary between populations and are often impaired in low income, rural settings. Drivers of these differences are not fully elucidated, hampering identification of strategies for optimising vaccine effectiveness. We hypothesise that urban–rural (and regional and international) differences in vaccine responses are mediated to an important extent by differential exposure to chronic infections, particularly parasitic infections.

**Methods and analysis** Three related trials sharing core elements of study design and procedures (allowing comparison of outcomes across the trials) will test the effects of (1) individually randomised intervention against schistosomiasis (trial A) and malaria (trial B), and (2) Bacillus Calmette-Guérin (BCG) revaccination (trial C), on a common set of vaccine responses. We will enrol adolescents from Ugandan schools in rural high-schistosomiasis (trial A) and rural high-malaria (trial B) settings and from an established urban birth cohort (trial C). All participants will receive BCG on day '0'; yellow fever, oral typhoid and human papilloma virus (HPV) vaccines at week 4; and HPV and tetanus/diphtheria booster vaccine at week 28. Primary outcomes are BCG-specific IFN-γ responses (8 weeks after BCG) and for other vaccines, antibody responses to key vaccine antigens at 4 weeks after immunisation. Secondary analyses will determine effects of interventions on correlates of protective immunity, vaccine response waning, priming versus boosting immunisations, and parasite infection status and intensity. Overarching analyses will compare outcomes between the three trial settings. Sample archives will offer opportunities for exploratory evaluation of the role of immunological and 'trans-kingdom' mediators in parasite modulation of vaccine-specific responses.

**Ethics and dissemination** Ethics approval has been obtained from relevant Ugandan and UK ethics committees. Results will be shared with Uganda Ministry of Health, relevant district councils, community leaders and study participants. Further dissemination will be done through conference proceedings and publications.

### Strengths and limitations of this study

► This will be the first well-powered programme of work to investigate effects of schistosomiasis treatment, of malaria treatment, and of Bacillus Calmette-Guérin revaccination on vaccine responses in adolescents.

► A major strength of this work is the opportunity to synthesise findings from three different study settings with differential parasite exposure using causal mediation analysis to obtain a comprehensive understanding of how parasitic infections influence vaccine responses in human populations.

► The results will provide insight into effects of parasites on infectious disease susceptibility: immunisation, notably with live vaccines, offers a surrogate for infection challenge in human subjects.

► The sample archives developed will provide a major asset for exploration of new leads arising from this hypothesis-driven work, or for an alternative, 'systems biology' approach investigating, for example, transcriptome, microbiome and virome.

► One limitation is that observational analyses of parasite effects are subject to potential unmeasured confounding; this will be mitigated by cautious interpretation of results and our intervention studies will address causality rigorously.

**Trial registration numbers** ISRCTN60517191, ISRCTN62041885, ISRCTN10482904.

## INTRODUCTION
### Population differences in vaccine responses

Effective vaccines are key weapons against infectious diseases,[1] but are still lacking for many poverty-related, neglected, emerging and re-emerging infections. Vaccine responses vary between populations and are often impaired in low income, rural settings.[2–6] A notable example is Bacillus Calmette-Guérin (BCG): both vaccine response and efficacy against tuberculosis differ internationally[3 4] and regionally.[6 7] Among other vaccines, yellow fever vaccine induced lower neutralising antibody levels, and responses waned faster, in Uganda compared with Switzerland.[5] Oral rotavirus and polio vaccines are also affected.[2] Within country, influenza and tetanus responses differed between urban and rural Gabon.[8 9] Responses to candidate tuberculosis,[10] malaria[11] and Ebola[12] vaccines are lower in Africa than in Europe or America. Prior exposure to the pathogen targeted by the vaccine, or to related organisms, may contribute to this phenomenon, but recent analyses implicate broader 'environmental sensitisation',[6] the drivers of which have not been determined. Prior exposure cannot explain results for vaccines against rare organisms, such as Ebola. Thus, drivers of POPulation differences in VACcine responses (POPVAC) are not fully elucidated; improved understanding is important for effective vaccine development and implementation.

The POPVAC programme comprises three trials, A, B and C, designed to address this challenge. The individual trial protocols are presented separately in this journal (bmjopen-2020-040426, bmjopen-2020-040427 and bmjopen-2020-040430). This 'Protocol X' provides an overview of our hypotheses and objectives.

### Immunomodulation by parasitic infections

Parasitic infections are important in tropical low-income countries (LICs),[13–15] and long proposed as modulators of vaccine responses.[16–19] As detailed in POPVAC A (bmjopen-2020-040426) and POPVAC B (bmjopen-2020-040427) protocols, animal models[20–22] and observational human studies[23–25] support this hypothesis, but no well-powered trials have been conducted to evaluate causality and reversibility of parasite effects on vaccine responses in adolescents or adults.[26]

### Trained immunity

Exposure to other unrelated infections or microbial antigens may also contribute to 'environmental sensitisation', modulating vaccine responses through training of the innate immune system.[27] BCG immunisation is a key model for this effect,[28 29] as considered in Protocol C (bmjopen-2020-040430).

### The 'transkingdom' concept

The 'transkingdom' concept[30] emphasises that mammals support a complex ecosystem of multicellular organisms, such as helminths, as well as bacteria, fungi, protozoa and viruses, and suggests that these interact in their effects on the mammalian immune system, rather than acting alone, as individual agents.[30] For example, in a mouse model, infection with the gut helminth *Heligmosomoides polygyrus*, or exposure to schistosome eggs, activated latent herpesvirus infection via alternative macrophage activation and the IL-4/Stat6 pathway.[31] In a contrasting study, *H. polygyrus* caused enhanced responses to respiratory syncytial virus in the mouse lung through interaction with the gut microbiome, translocation of microbial products to the circulation and enhanced systemic type I interferon expression.[32] Interestingly, in these studies, the same helminth resulted in opposite outcomes for latent viruses (for which it induced activation), versus exogenous viruses (for which it improved control).

Little has been done to evaluate these phenomena in human populations but, regarding latent herpesviruses our studies show that parasite exposure associates with elevated Kaposi's Sarcoma Herpesvirus antibody prevalence and titre (indicating viral activation).[33–35] The impact of malaria on Epstein-Barr virus, promoting induction of Burkitt's lymphoma, is well recognised.[36] Cytomegalovirus has major immunological effects, including impact on vaccine responses.[37]

In humans, evidence of enhanced microbial translocation (MT) has been found during *Schistosoma mansoni*,[38] hookworm[39] and *Strongyloides*[40] infection, with altered expression of parameters such as toll-like receptor expression, but without the level of immune activation associated with septic shock or with MT in HIV infection. Nevertheless, the second mouse model discussed above shows that this may have profound effects on responses to infectious agents at remote sites.[32]

Thus, herpesvirus activation and, or, MT may mediate, in part, parasite-induced modulation of vaccine responses.

### Differences in immunological characteristics between populations

Whatever the key exposures and mechanisms involved, immunological characteristics differ markedly between populations internationally,[5 41 42] and between urban and rural settings.[41 43 44] Characteristics that differ include gene methylation and expression (not solely attributable to population genetics)[43 45]; responses to innate stimuli[42 45]; frequency and activation of innate immune cells, T and B cells, and memory cell pools.[5 41] Understanding the immunological predictors of vaccine response, and factors that drive them, will contribute to strategies for improving vaccine efficacy for rural, tropical settings.

## HYPOTHESIS AND OBJECTIVES

The overarching goal of the 'POPVAC' is to understand POPVAC, in order to identify strategies through which vaccine effectiveness can be optimised for low income, tropical settings where they are especially needed. We

**Figure 1** Hypothesised pathways to population differences in vaccine responses.

focus on the hypothesis (figure 1) that geographical differences in vaccine responses are mediated to an important extent by differential exposure to chronic, particularly parasitic, infections; that parasites act in part via 'transkingdom' effects; and that these exposures impact the preimmunisation immune profile and hence vaccine response (and efficacy).

We will address this hypothesis in three, linked trials (POPVAC A, B and C; detailed protocols published separately in this journal: bmjopen-2020-040426, bmjopen-2020-040427 and bmjopen-2020-040430, respectively) which share core elements of study design and procedures, allowing comparison of outcomes across the trials. Each trial will test effects of a different randomised intervention on a common set of vaccine responses. POPVAC A will determine the effect of intensive schistosomiasis treatment on vaccine responses among rural island adolescents. POPVAC B will determine the effect of intensive malaria treatment on vaccine responses among rural

adolescents. POPVAC C will determine the effect of BCG revaccination on responses to unrelated vaccines.

This paper describes background and methods common to all three trials, summarises objectives linking the trials, and details planned approaches to cross-cutting objectives which will use data and samples from all trials. Standard Protocol Items: Recommendations for Interventional Trials reporting guidelines[46] are used.

Our objectives are to

1. Determine whether there are reversible effects of chronic parasitic infection on vaccine response (POPVAC A and B).
2. Determine whether BCG 'preimmunisation' enhances responses to unrelated vaccines (POPVAC C).
3. Determine which life-course exposures influence vaccine responses in adolescence (using data from POPVAC C).
4. Compare vaccine response profiles between three Ugandan settings (using data from all trials): rural,

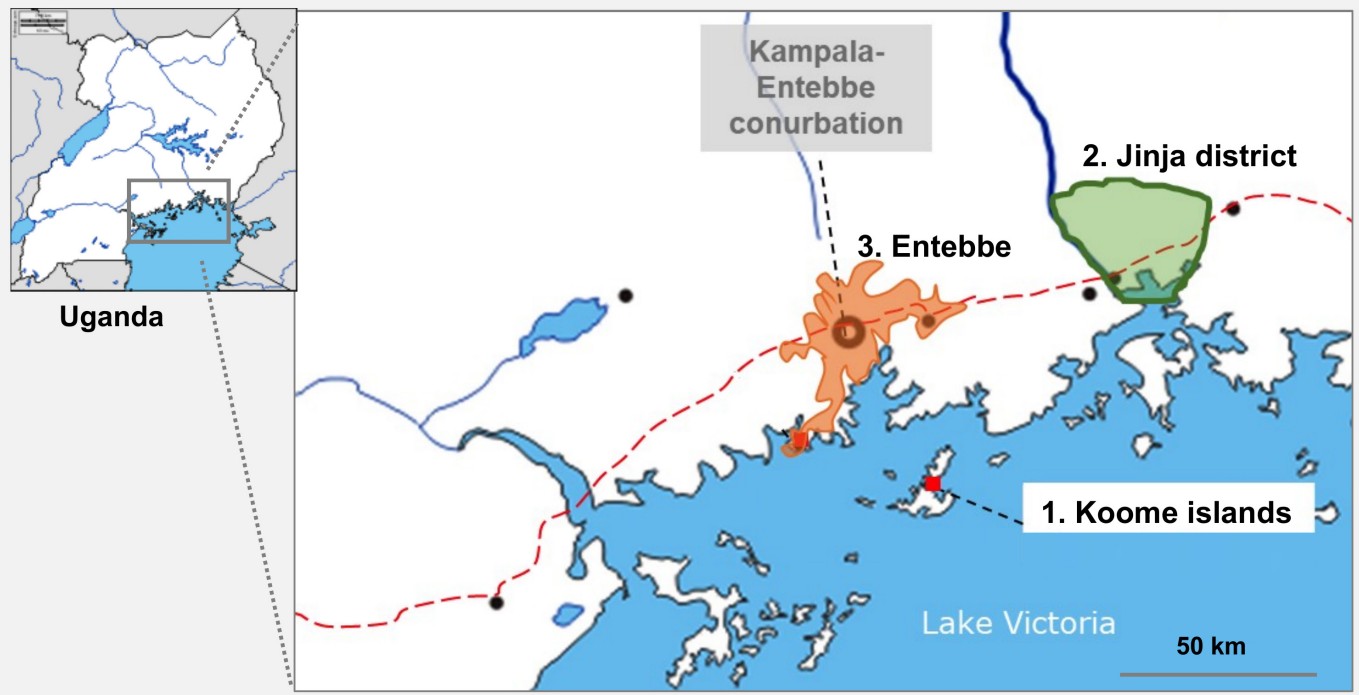

**Figure 2** Study sites.

high schistosomiasis exposure; rural, high malaria exposure and urban.

5. Explore the role of 'transkingdom' interactions in determining vaccine responses (using samples and data from all Trials).

6. Investigate preimmunisation immunological parameters associated with vaccine responses and determine whether these are driven by parasite or microbial exposure (using samples and data from all trials).

## METHODS AND ANALYSIS
### Setting
Uganda still experiences high schistosomiasis[47 48] and malaria burdens.[49–51] Our study sites (figure 2) will be Lake Victoria Koome Islands (high schistosomiasis),[47 52] Jinja district rural subcounties (high malaria)[51] and Entebbe (urban; low schistosomiasis and low malaria).[52] These settings provide ideal opportunities to investigate effects of schistosomiasis and malaria on vaccine responses. Geohelminths are less common in our settings[47 53]: hookworm, especially, has declined dramatically following government intervention programmes; therefore, geohelminths will be considered as potential confounders, where appropriate, but not prioritised.

### Cohorts
This work will involve adolescents from two rural settings and one urban cohort (figure 3). Rural trials will recruit participants aged 9–17 years from primary schools in Koome islands (POPVAC A) and Jinja district (POPVAC B), selected purposefully to assure schistosomiasis and malaria prevalence appropriate to our design. This will

allow us to investigate our two major infections separately, averting the risk that a potent effect of one infection masks an impact of the other, or of its treatment.

POPVAC C will recruit members of the Entebbe Mother and Baby Study (EMaBS) birth cohort[54]: 2500 women were recruited between 2003 and 2005 in a trial of anthelminthic treatment during pregnancy, investigating effects on infants' vaccine responses.[54] Children from the EMaBS birth cohort will be aged 13 to 17 years during recruitment to this study; about 300 individuals are expected to take part.

### Interventions
In the high-schistosomiasis cohort (trial A), we will individually randomise participants to intensive or standard praziquantel (PZQ) treatment, in a 1:1 ratio, in an open-label, parallel group trial.

In the high-malaria cohort (trial B), we will individually randomise participants to monthly dihydroartemisinin piperaquine (DP) versus placebo in a double-blind, placebo-controlled trial.

In the urban (EMaBS) cohort (trial C), we will individually randomise participants to BCG revaccination, or no BCG revaccination, as the first component of the vaccine schedule, in a controlled, open-label, parallel-group trial.

Recruitment criteria, interventions, randomisation and treatment allocation procedures are detailed in protocols for trials A, B and C (bmjopen-2020-040426, bmjopen-2020-040427 and bmjopen-2020-040430, respectively; published in this journal).

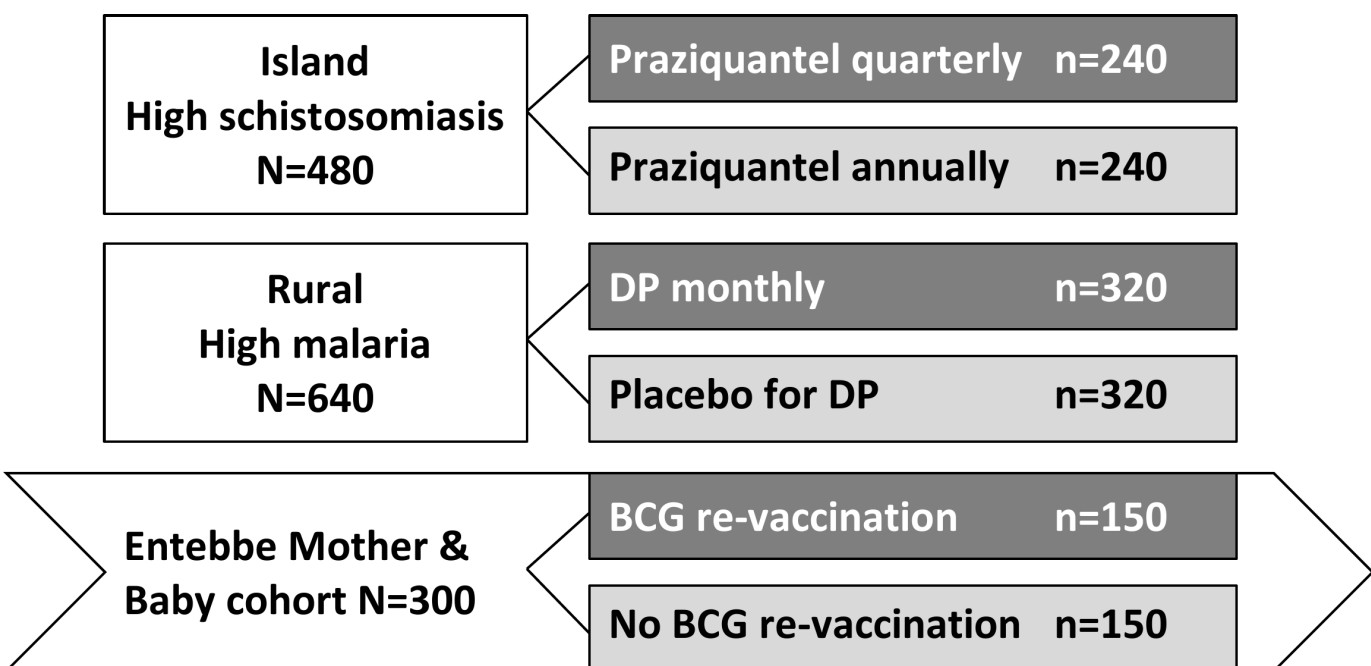

**Figure 3** Overall programme design and sample sizes. BCG, bacillus calmette-guérin; DP, dihydroartemisinin.

## Immunisations

We have previously highlighted the complexity of helminth effects on vaccine responses.[26] Differences in parasite effects on live and non-live, oral and parenteral, priming and boosting vaccines may contribute to this. Activated innate responses may kill live vaccines and suppress subsequent adaptive responses,[5 55] but bias, or enhance, responses to toxoids or proteins[17 23 56]; intestinal inflammation may impair responses to live oral vaccines[57]; priming may be more vulnerable than boosting.[58 59] Thus, results from a single-vaccine study would not be generalisable.

Therefore, we will study a portfolio of licensed vaccines expected to be beneficial (some already given) to adolescents in Uganda: live parenteral (BCG, yellow fever), live oral (typhoid) and non-live (HPV) (a viral particle vaccine) and tetanus/diphtheria (Td; toxoid vaccines).

This will allow us to compare effects of interventions and exposures between vaccine types. Each cohort will receive the same vaccine portfolio (table 1), comprising three main immunisation days (weeks 0, 4 and 28). Additional HPV immunisation will be provided for girls aged 14 years or above, and a second Td boost will be given after study completion, to accord with national Expanded Programme on Immunisation (EPI) routines, but responses to these will not specifically be addressed. Further rationale for vaccine selection is detailed in online supplemental material 1. Our schedule has been developed in consultation with the EPI programme (see online supplemental table 1) and is cognizant of potential interference between vaccines (see online supplemental material 1, online supplemental table 2).

Although optimal timings for outcome measures vary between vaccines, sampling (for primary endpoints)

| Table 1 | Immunisation schedule | | | | |
|---|---|---|---|---|---|
| | Immunisation week 0 | Immunisation week 4 | (Immunisation week 8) | Immunisation week 28 | (Immunisation week 52) |
| Live vaccines | BCG vaccination/ revaccination*, †, ‡ | Yellow fever (YF-17D) Oral typhoid (Ty21a) | | | |
| Non-live vaccines | | HPV prime | HPV boost for girls aged ≥14 years§, ¶ | HPV boost and tetanus/diphtheria boost | Tetanus/diphtheria boost¶, ** |

*All participants in the urban (Entebbe Mother and Baby Study, EMaBS) cohort received BCG at birth, so within EMaBS this will berevaccination; prior BCG status may vary for the rural cohorts (data on history and documentation of prior BCG, and presence of a BCG scar, will be documented although these approaches have limitations for determining BCG status).
†EMaBS participants will be randomised to receive BCG 'preimmunisation' or not as part of trial C.
‡All participants in rural cohorts will receive BCG (trials A and B).
§The National EPI programme recommends three doses of HPV vaccine for older girls.
¶These doses will be given to comply with guidelines but outcomes specifically relating to these doses will not be assessed.
**Priming by immunisation in infancy is assumed.
BCG, bacillus calmette-guérin; HPV, human papilloma virus.

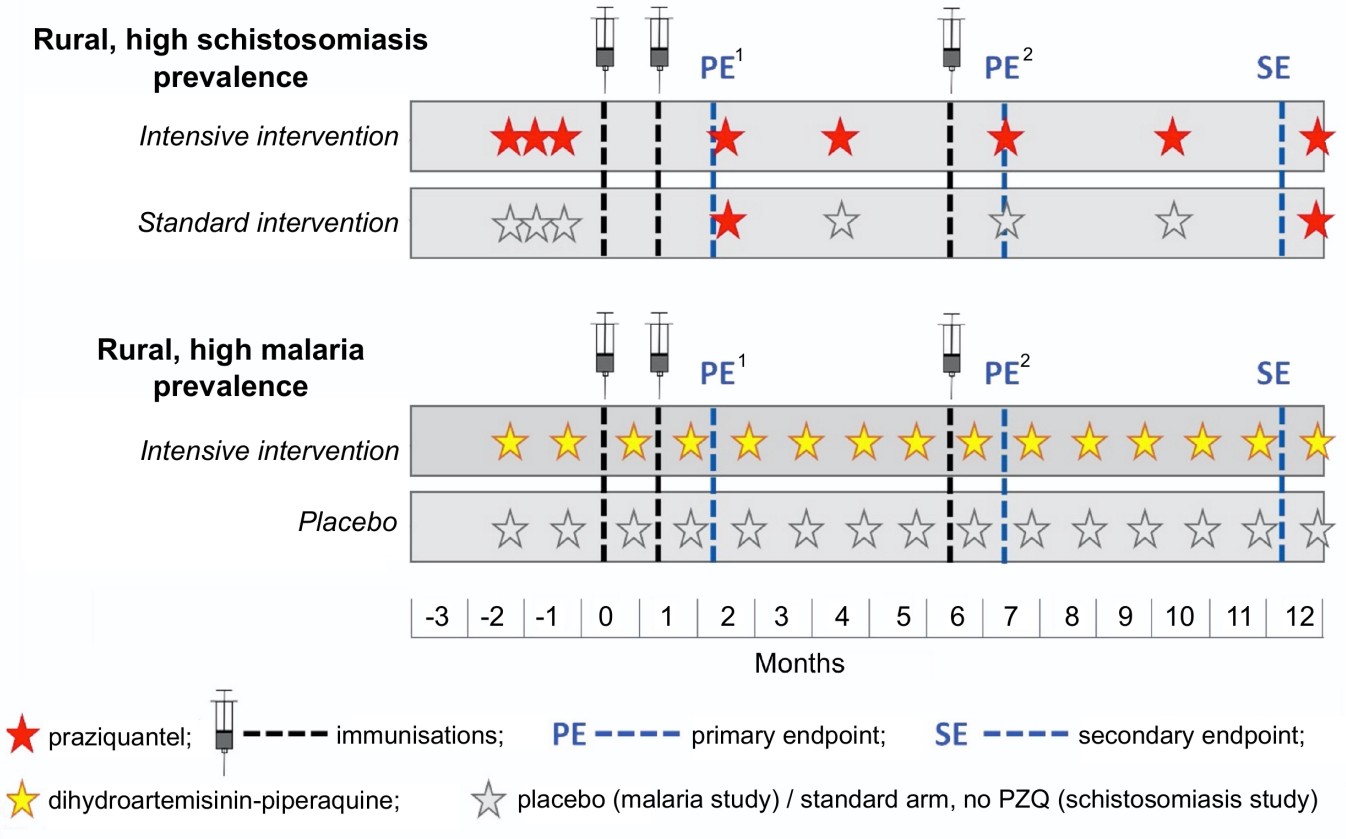

**Figure 4** Outline of immunisations and interventions.[1]Primary endpoints will be at 8 weeks post-BCG and 4 weeks postyellow fever (YF-17D), oral typhoid (Ty21a), human papilloma virus and tetanus/diptheria vaccination.[2]Primary endpoint for responses to td given at 28 weeks. BCG, Bacillus Calmette-Guérin.

will be done at 8 weeks post BCG, 4 weeks post YF-17D, Ty21a, HPV and Td (figure 4), targeting establishment of memory responses and antibody response peaks. A secondary endpoint at 1 year will assess waning. Analyses will take baseline measurements into account.

### Outcomes

Primary outcomes, assessed in all participants, will be

1. BCG: BCG-specific IFN-γ ELISpot response 8 weeks post-BCG immunisation: this response is associated with decreased risk of tuberculosis disease postimmunisation in infants.[60]
2. YF-17D: neutralising antibody titres (plaque-reduction neutralisation test) 4 weeks post-YF immunisation.
3. Ty21a: Salmonella typhi lipopolysaccharide-specific IgG concentration 4 weeks post-Ty21a immunisation.
4. HPV: IgG specific for L1-proteins of HPV-16/18 4 weeks post-HPV priming immunisation.
5. Td: tetanus and diphtheria toxoid-specific IgG concentration 4 weeks post-Td immunisation.

Secondary outcomes, assessed in all participants, will further investigate estimates of protective immunity (for vaccines where these are available) and dynamics of the vaccine responses, as well as the impact of interventions on parasite clearance.

1. *Protective immunity.* Proportions with protective neutralising antibody (YF); protective IgG levels (TT)[61]; seroconversion rates (Ty21a) 4 weeks postimmunisation.
2. *Response waning.* Primary outcome measures (all vaccines) repeated at week 52, and area under the curve analyses. Parasites may accelerate,[58] and interventions delay, waning.
3. *Priming versus boosting.* Effects on priming versus boosting will be examined for HPV, comparing outcomes 4 weeks after the first and second vaccine doses.

Our sample collection will offer opportunities for an array of exploratory immunological evaluations, focusing on vaccine antigen specific outcomes. Exploratory assessments will provide detail on immune response characteristics over the study time-course, and on the role of immunological profiles and trans-kingdom effects in mediating modulation of vaccine-specific responses.

### Sample size

Our sample size estimates focus on primary outcomes for the main comparisons for objectives i, ii and iv.

Based on literature,[5 60 62] we anticipate SDs of primary outcome measures lying between 0.3 and 0.6 $\log_{10}$; responses in rural, high-parasite settings 0.3–0.4 $\log_{10}$ smaller than in the urban setting[63 64] and effective treatment restoring responses by approximately 0.2 $\log_{10}$.[63] We, therefore, power our study to detect differences of this magnitude (0.2 $\log_{10}$) or smaller. We assume *S. mansoni* prevalence of ≥80% in the high-schistosomiasis setting[47] and malaria infection prevalence of ≥60% in the high-malaria setting.

Our planned sample sizes are as follows:

High-schistosomiasis setting (objective i, trial A): 480 (240 quarterly PZQ, 240 annual PZQ); of whom we anticipate 384 will be *S. mansoni* infected, giving 192 participants in each trial arm with *S. mansoni* infection at baseline.

High-malaria setting (objective i, trial B): 640 (320 DP, 320 placebo) of whom we anticipate 384 will be malaria infected, giving 192 participants with malaria in each trial arm at baseline.

Urban setting (objective ii, trial C): 300 EMaBS participants (150 BCG 'preimmunisation', 150 no BCG immunisation).

Urban versus rural and urban versus island comparisons (objective iv): 150 urban EMaBS, 240 rural high-schistosomiasis control group participants and 320 rural high-malaria control group participants will be included. Allowing 20% lost to follow-up in rural cohorts and 10% in the EMaBS cohort, this will give >80% power to detect a difference of 0.14$\log_{10}$ or more in vaccine responses in urban compared with each rural setting at 5% significance level assuming vaccine response SD of 0.4$\log_{10}$.

Table 2 shows power estimates for objective i, ii and iv. Sample size considerations for additional analyses are detailed in the protocol papers for the individual trials (bmjopen-2020-040426, bmjopen-2020-040427 and bmjopen-2020-040430).

## Approach to trial objectives

Approaches to objective i (trials A and B) and objective ii (trial C) are detailed in the focused papers for these trials (bmjopen-2020-040426, bmjopen-2020-040427 and bmjopen-2020-040430). Here we present approaches to objectives iii–vi.

### Objective iii: life course exposures that influence vaccine responses in adolescence (Trial C)

Data collected in the EMaBS over the participants' life course (and as part of this protocol) will be used in regression analyses to investigate associations between infection exposure in utero, infancy, childhood and adolescence on vaccine responses. Sociodemographic variables will be considered as potential confounders. We will include the following important variables: age, sex, sociodemographic variables, BCG strain received at birth (and other vaccines received), helminth-related exposures, malaria and documented illness events.

Regression analyses will be used to evaluate associations between infections and outcomes, with adjustment for confounders. We will use a hierarchical statistical modelling approach, so that for analysis of early life exposures we will not adjust for later life exposures that may be on the causal pathway, but for associations

| Table 2 | Power estimates for objectives I, II and iv (5% significance level) | | | | | | |
|---|---|---|---|---|---|---|---|
| | **Log₁₀ difference** | | | | | | |
| **SD (log₁₀)** | **0.08** | **0.10** | **0.12** | **0.14** | **0.16** | **0.18** | **0.20** |
| Objective i: 192 high intensity vs 192 low intensity (infected only) | | | | | | | |
| 0.3 | 65% | 83% | 94% | 98% | >99% | >99% | >99% |
| 0.4 | 42% | 59% | 75% | 87% | 94% | 98% | 99% |
| 0.5 | 29% | 42% | 56% | 69% | 80% | 88% | 94% |
| 0.6 | 21% | 31% | 42% | 53% | 65% | 75% | 83% |
| Objective ii: 150 BCG 'preimmunisation' vs 150 no BCG vaccination | | | | | | | |
| 0.3 | 59% | 78% | 91% | 97% | 99% | >99% | >99% |
| 0.4 | 37% | 53% | 69% | 82% | 91% | 96% | 98% |
| 0.5 | 26% | 37% | 50% | 63% | 75% | 84% | 91% |
| 0.6 | 19% | 28% | 37% | 48% | 59% | 69% | 78% |
| Objective iv: 240 rural vs 150 urban* | | | | | | | |
| 0.3 | 66% | 84% | 94% | 99% | >99% | >99% | >99% |
| 0.4 | 43% | 60% | 76% | 87% | 94% | 98% | 99% |
| 0.5 | 30% | 43% | 57% | 80% | 81% | 89% | 94% |
| 0.6 | 22% | 32% | 43% | 54% | 66% | 76% | 84% |

Cells highlighted in grey correspond to >80% power.
*Numbers shown for rural high schistosomiasis versus urban setting. Power will be greater for rural high malaria versus urban setting.
BCG, bacillus calmette-guérin.

between later childhood or current exposures and vaccine responses we will adjust for early life exposures as potential confounders. Linear regression will be used for the primary outcomes and for continuous secondary outcomes. Outcome distributions are likely to be positively skewed; where necessary, we will apply log transformations to normalise outcome distributions before linear regression analysis. Logistic regression will be used for the protective immunity secondary outcomes, which are binary. We shall also investigate whether multiple infection exposures combine multiplicatively in their effect and test for interaction.

Genetic factors will also be considered: genetic data is already available for EMaBS based on earlier approvals for work on genetic polymorphisms and vaccine responses in infancy. It will be of interest to determine whether genetic factors also have a strong influence on adolescents' responses.

### Objective iv: urban–rural comparisons in vaccine response

We hypothesise that environmental (especially parasite) exposures are key drivers of POPVAC. Hence, we predict differences between urban and rural settings within Uganda (with urban vaccine responses stronger, as observed in Gabon[8 9] and Senegal[64] and that these differences will be related to parasite exposure.

The key exposure for this objective is 'setting'. We will compare outcomes between urban EMaBS and (1) rural high-schistosomiasis and (2) rural high-malaria participants. For these comparisons we shall include in the analysis only the urban EMaBS participants who were randomised to receive BCG 'preimmunisation,' such that their immunisation schedule is identical to that in the other two settings. We will include only control groups from the rural high-schistosomiasis and high-malaria settings, since these will have received minimal anti-parasite treatment (figures 3 and 4). In the primary analysis, we will adjust for age and sex, but not for factors likely to be on the causal pathway between setting and vaccine response.

Although our settings are purposely chosen based on parasite prevalence, there will be overlap in this, and other, exposures. Therefore, we will undertake exploratory analyses using causal mediation modelling, a statistical approach which aims to identify factors that mediate an observed association[65] (here, between setting and vaccine response). We will focus on current *S. mansoni and Plasmodium falciparum* infection, and prior exposure (assessed by anti-schistosome and antimalaria antibody), as key potential mediators of interest (see online supplemental figure 1). Explanatory factors for urban versus rural setting, and for vaccine responses, will be included in the causal diagram and adjusted for in analyses. Although nutrition may impact vaccine responses,[66] we expect this to be less important in healthy adolescents than might be the case in young children; however, data on anthropometric parameters and diet will be collected and adjusted for.

One limitation is that observational analyses of parasite effects are beset with potential unmeasured confounding factors. Results will be interpreted cautiously and objective i (trials A and B) will address causality rigorously. The sample archive will allow future investigation of additional potential mediators. Another potential limitation is that in Uganda, rural to urban migration for schooling is common. However, in the EMaBS birth cohort we have data on residence. Urban to rural migration for schooling is relatively unlikely but will be documented.

With objective iv, we expect to confirm differences in vaccine responses between settings; and to obtain insights into environmental factors that mediate this. By addressing the portfolio of vaccines and array of outcomes presented above, we will identify categories of vaccine most affected; we will distinguish effects on waning and (for HPV) on priming versus boosting. Vaccine response data from urban EMaBS adolescents will provide a reference, suggesting the extent to which changes in infection exposure and lifestyle are likely to influence responses to vaccines (and to infectious diseases) as Africa's urbanisation advances.

### Objective v: the role of 'transkingdom' interactions in determining vaccine responses

We will address the hypothesis that parasites also impact vaccine responses through 'transkingdom' effects, mediated by other components of the host ecosystem. We propose that herpesvirus activation and MT are likely to be determinants of vaccine response, and they are driven by parasitic infections. Samples from selected participants from each setting and study arm will be examined. To optimise the precision of our comparisons these will be selected among those (for the high-schistosomiasis and high-malaria settings) who had the parasitic infection of interest at baseline (and, if possible, no other parasitic infection detected) and complied with the intended treatment. To test our hypothesis, we will measure markers of viral activation and MT in plasma/serum and/or stool. Initial analyses (figure 5) will investigate associations between these markers and vaccine responses (arrow F); and between settings (arrow D) or parasite infection and treatment (arrow E) and viral activation or MT. Objective vi will link herpesvirus activation, MT and parasitic infections to immune activation and regulation.

### Objective vi: preimmunisation immunological parameters

There is strong evidence that preimmunisation immunological status impacts vaccine responses[5 60] but the factors that determine this status have not been identified. We hypothesise that the environmental, parasitic, viral and microbial exposures addressed in objectives i–v are key (figure 5). We aim to investigate this by identifying immunological parameters that specifically link the distal exposures to vaccine response. These include the circulating cytokine and chemokine milieu, innate cell responses (which govern adaptive responses), and frequencies and phenotypes of both innate and adaptive cells. Our

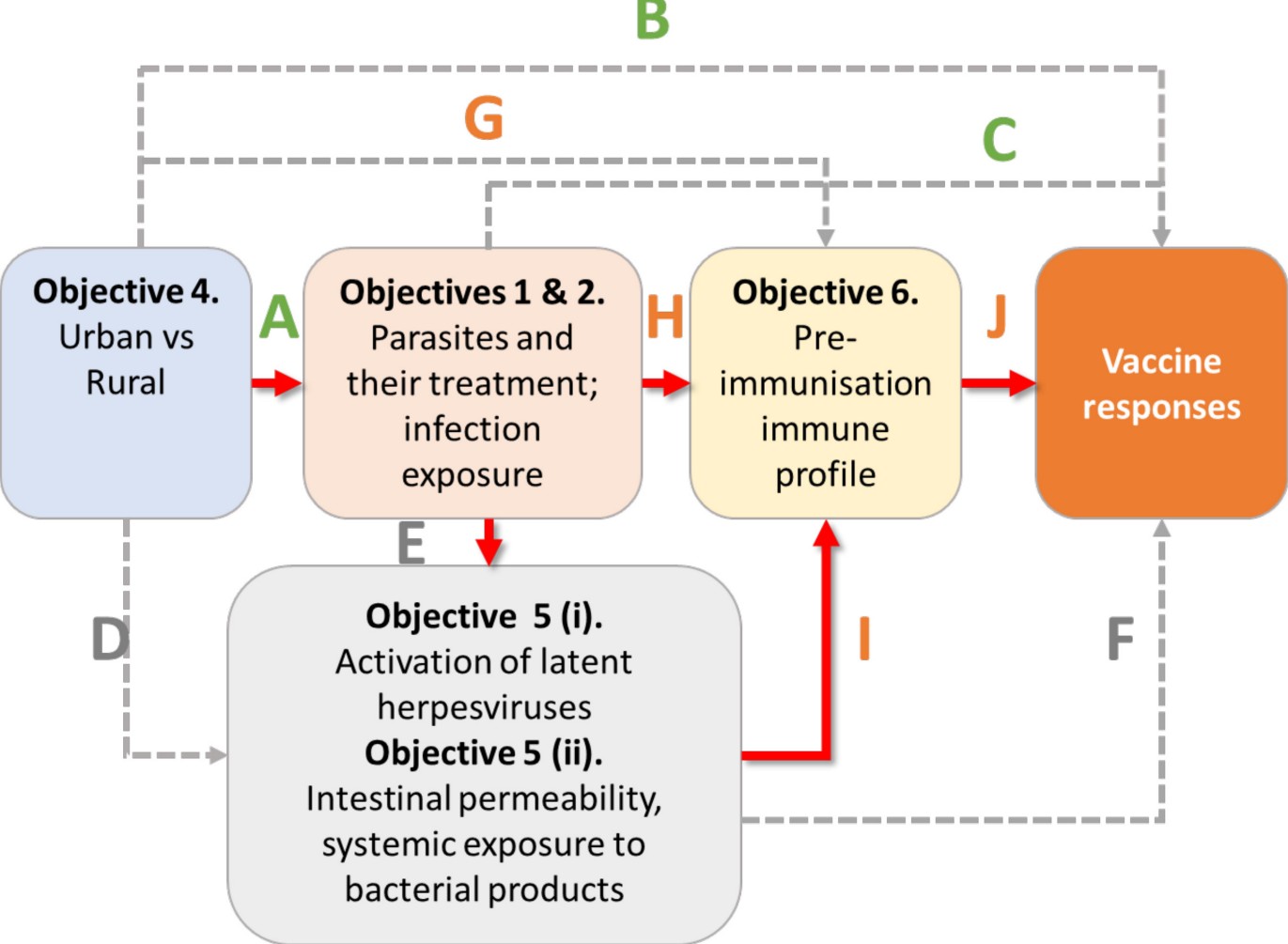

**Figure 5** Synthesised analysis of study objectives. Red arrows represent our principal hypotheses. Arrows A–C are considered in objectives I and iv; Arrows D–F in objective v; Arrows G–J, in objective vi; as well as the fully synthesised analysis.

principal focus will be on pre-immunisation measurements although, in selected groups of most interest, it will also be possible to examine samples obtained post-immunisation to identify which biomarkers or cell populations change.

Initial analyses will confirm hypothesised, and identify new, preimmunisation immunological parameters associated with responses to vaccines in our portfolio (figure 5, arrow J). If we observe associations between rural versus urban location (arrow B) or parasites and their treatment (arrow C) and vaccines responses, we will explore whether these effects are mediated by viral and MT variables and preimmunisation immune profiles using causal mediation analyses.[65]

### Operational considerations

A programme steering committee has been set up to guide progress across all projects. A data and safety monitoring board has also been appointed to provide real-time safety oversight. Details of these and other operational activities can be found in online supplemental material 1.

### Ethics and dissemination

Ethical and regulatory approval has been granted from the Research Ethics Committees of Uganda Virus Research Institute and the London School of Hygiene and Tropical Medicine, the Uganda National Council for Science and Technology and the Uganda National Drug Authority; details are given in online supplemental material 1 and in each trial paper. Given the importance of the data and sample archive as a resource for mechanistic studies on the determinants of vaccine responses, assent and consent processes will include storage of samples for future and genetic studies, and anonymised data and sample sharing. Any protocol amendments will be submitted to ethics committees and regulatory bodies for approval before implementation.

Study findings will be published through open access peer-reviewed journals, presentations at local, national and international conferences and to the local community through community meetings. Anonymised participant level datasets generated will be available on request.

## Patient and public involvement

Concepts involved in this work have been discussed with colleagues at the Vector Control Division and EPI in the Ministry of Health (Uganda) and with relevant District Councils, community leaders and Village Health Teams. We also have held meetings to explain the proposed work to teachers, parents, participants and village members, and to address their questions. Study findings will be shared with these stakeholders.

## Data management and analysis

Sociodemographic information and clinical and laboratory measurements will be recorded and managed using Research Electronic Data Capture tools,[67 68] with paper-based forms as back-up. All data will be recorded under a unique study ID number. When paper forms must be used, data will be entered in a study-specific database, with standard checks for discrepancies. All data for analysis will be anonymised and stored on a secure and password-protected server, with access limited to essential research personnel. Anonymised participant-level datasets generated will be available for sharing on request.

## DISCUSSION

This will be the first well-powered set of studies to investigate effects of schistosomiasis and malaria treatment, and of BCG revaccination, on vaccine responses in adolescents. The results will add to understanding of POPVAC and of interventions that may enhance them. The sample archives developed will provide a major asset for exploration of new leads arising from this hypothesis-driven work, or for an alternative, 'systems biology' approach investigating, for example, transcriptome, microbiome and virome.

Our focus is on the immunological effects of infection exposure in human participants. By understanding these effects, we aim to inform and promote vaccine design tailored to the challenging environment of LICs, and to inform the development of public health strategies (such as tailored immunisation regimens and combined parasite-control/immunisation interventions) that will optimise vaccine implementation in parasite-endemic settings.

Our strong immunoepidemiological design and nested immunological studies will address specific hypotheses regarding pathways of effects. Population immunology is useful for translation of findings to and from basic (especially animal) studies to human health. Our randomised design will determine causal, and reversible, effects of parasitic infections. Substantial sample sizes are needed because immune responses are highly variable in human populations.

We have several reasons for studying vaccine responses particularly among adolescents. In this study setting, they bear a heavy parasite burden.[69] As well, this age group is a target group for vaccines against tuberculosis and sexually transmitted infections (currently HPV—in future,

it is hoped, for vaccines against HIV) and for booster immunisations. Also, they enter a period of increased risk of pulmonary tuberculosis after the relatively low-risk period of mid-childhood, and are thus a target group for improved vaccines for tuberculosis.

## Study timeline

POPVAC A began recruiting in July 2019. Intervention will be up to 12 months, with completion of the project scheduled for September 2020. POPVAC B is scheduled begin recruiting in February 2021. Intervention will be up to 12 months, with completion of the project scheduled for April 2022. POPVAC C is scheduled to begin recruiting in May 2020. Intervention will be up to 12 months, with completion of the project scheduled for April 2022.

**Author affiliations**

[1]Immunomodulation and Vaccines Programme, Medical Research Council/Uganda Virus Research Institute and London School of Hygiene and Tropical Medicine (MRC/UVRI and LSHTM) Uganda Research Unit, Entebbe, Uganda
[2]Uganda National Expanded Program on Immunisation, Ministry of Health, Kampala, Uganda
[3]Department of Clinical Research, London School of Hygiene and Tropical Medicine, London, London
[4]MRC Tropical Epidemiology Group, Department of Infectious Disease Epidemiology, London School of Hygiene and Tropical Medicine, London, UK

**Acknowledgements** We thank the Uganda National Expanded Programme for Immunisation, the Vector Control Division of the Ministry of Health, the Ministry of Education, Jinja district local government, Wakiso district local government and Entebbe hospital for their support. We thank members of the POPVAC programme steering committee (chaired by Professor Richard Hayes) and the Data and Safety Monitoring Board (Dr David Meya, Professor Andrew Prendergast and Dr Elizabeth George).

**Collaborators** Principal investigator: Alison Elliott; Project leader: Ludoviko Zirimenya; laboratory staff: Gyaviira Nkurunungi, Stephen Cose, Rebecca Amongin, Beatrice Nassanga, Jacent Nassuuna, Irene Nambuya, Prossy Kabuubi, Emmanuel Niwagaba, Gloria Oduru, Grace Kabami; statisticians and data managers: Emily Webb, Agnes Natukunda, Helen Akurut, Alex Mutebe; clinicians: Anne Wajja, Milly Namutebi, Christopher Zziwa; nurses: Caroline Onen, Esther Nakazibwe, Josephine Tumusiime, Caroline Ninsiima, Susan Amongi, Florence Akello; internal monitor: Mirriam Akello; field workers: Robert Kizindo, Moses Sewankambo, Denis Nsubuga, Samuel Kiwanuka, Fred Kiwudhu; boatman: David Abiriga; administrative management: Moses Kizza, Samsi Nansukusa; internal and external collaborators: Pontiano Kaleebu, Hermelijn Smits, Maria Yazdanbakhsh, Govert van Dam, Paul Corstjens, Sarah Staedke, Henry Luzze, James Kaweesa, Edridah Tukahebwa, Elly Tumushabe, Moses Muwanga.

**Contributors** AME conceived the study. AME, GN, EW, AN, SC, LZ, JN, AW, PK and HL contributed to study design. LZ, GO, CN, CZ and FA are site clinicians/nurses/clinical laboratory technicians providing valuable input on clinical considerations of the intervention. RK heads the team of field workers handling the organisational integration of the intervention. MA is the study internal monitor. AN and EW are involved in organisation of the databases, trial randomisation, treatment allocation and drawing up of analytical plans. GN, LZ, AN, SC, EW and AME drafted the manuscript. All authors reviewed the manuscript, contributed to it and approved the final version.

**Funding** The POPVAC programme of work is supported by the Medical Research Council of the UK (grant number MR/R02118X/1). SC and JN are supported in part by the Makerere University-Uganda Virus Research Institute Centre of Excellence for Infection and Immunity Research and Training (MUII-plus). MUII-plus is funded under the DELTAS Africa Initiative. The DELTAS Africa Initiative is an independent funding scheme of the African Academy of Sciences (AAS), Alliance for Accelerating Excellence in Science in Africa (AESA) and supported by the New Partnership for Africa's Development Planning and Coordinating Agency (NEPAD Agency) with funding from the Wellcome Trust (grant 107743) and the UK Government. The

MRC/UVRI and LSHTM Uganda Research Unit is jointly funded by the UK Medical Research Council (MRC) and the UK Department for International Development (DFID) under the MRC/DFID Concordat agreement and is also part of the EDCTP2 programme supported by the European Union.

**Disclaimer** The study sponsor (London School of Hygiene and Tropical Medicine) and funders had no role in study design; collection, management, analysis, and interpretation of data; writing of the protocol; and the decision to submit the protocol for publication.

**Map disclaimer** The depiction of boundaries on this map does not imply the expression of any opinion whatsoever on the part of BMJ (or any member of its group) concerning the legal status of any country, territory, jurisdiction or area or of its authorities. This map is provided without any warranty of any kind, either express or implied.

**Competing interests** AME reports a grant from the Medical research Council, UK (POPVAC programme funding). The rest of the authors declare that they have no conflicts of interest.

**Patient consent for publication** Not required.

**Provenance and peer review** Not commissioned; externally peer reviewed.

**ORCID iD**
Gyaviira Nkurunungi http://orcid.org/0000-0003-4062-9105

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
