## [Reviewer comments · BMJ Open]

ARTICLE DETAILS

TITLE (PROVISIONAL)	Population differences in vaccine responses (POPVAC): scientific rationale and cross-cutting analyses for three linked, randomised controlled trials assessing the role, reversibility and mediators of immunomodulation by chronic infections in the tropics
AUTHORS	Nkurunungi, Gyaviira; Zirimanya, Ludoviko; Natukunda, Agnes; Nassuuna, Jacent; Oduru, Gloria; Ninsiima, Caroline; Zziwa, Christopher; Akello, Florence; Kizindo, Robert; Akello, Mirriam; Kaleebu, Pontiano; Wajja, Anne; Luzze, Henry; Cose, Stephen; Webb, Emily; Elliott, Alison

VERSION 1 – REVIEW

REVIEWER	Liana F Wait Princeton University, USA
REVIEW RETURNED	01-Aug-2020

GENERAL COMMENTS	Overall, a very well written and detailed study protocol on an important topic. Some specific comments and suggestions: Will you test for the actual presence of parasites before treatment and throughout the trials? Line 298 implies that this will be done, but I would recommend including this information (and the parasitological methods you plan to use) explicitly in the Methods section. If you don't already intend to test this, I would highly recommend that you consider it! Line 100 – It seems to me that “trans-kingdom interaction” is a synonym for the more-widely used term “co-infection”, or at best it describes interactions that occur during co-infections. The only difference I can think of is that co-infection usually refers to co-habitation and interaction between two infectious organisms (parasites and pathogens), where trans-kingdom interactions seems to refer to interactions between pathogens and components of the microbiome. However, there isn't a clear line between microbiome and pathogens/parasites (and the 2016 Pfeiffer and Virgin “trans-kingdom interaction” paper cited here includes protozoa and helminths in the “microbiome” category), so I still think the terms are essentially synonymous. With this in mind, the literature on co-infections is far richer than the “trans-kingdom interaction” studies cited here, and I would recommend acknowledging its existence and citing some of these papers. I also recommend that you use the term “co-infection” alongside “trans-kingdom interactions” in your paper so that interested parties can find and read this paper. Line 120 – this is very certain phrasing and I would suggest couching it in some uncertainty since (I think?) it's your hypothesis,
---

	and not something that's been otherwise proven. Line 178 – typo: anthelminthic should be anthelmintic. When I searched, this spelling was used 4 times in the manuscript. Line 199 – this is excellent, we sorely need more studies investigating the effects of parasites on different types of vaccine and on vaccines with different routes of administration. (In my recent meta-analysis on parasite-vaccine interactions I was unable to test for any patterns due to route of administration because of a lack of studies that used non-parenterally administered vaccines.) Line 276 – typo (“infectionss”) Line 306 – sentence ends abruptly – should this say "confounding factors"?
--	---

REVIEWER	Simeon Cadmus University of Ibadan, Ibadan, Nigeria
REVIEW RETURNED	02-Aug-2020

GENERAL COMMENTS	This work is properly conceived and well written. It took into consideration several population studies with diverse inferences across the globe, and more fundamentally, explores the nuances therein to set the stage for this work. As stated by the authors, "understanding the immunological predictors of vaccine response, and factors that drive them, will contribute to strategies for improving vaccine efficacy for rural, tropical settings". This can not be overemphasized judging from the fact that there are a plethora of infectious diseases afflicting people in sub-Saharan African where issues related to weak health infrastructure further undermines the ability to solve this huge public health debacle. Going forward, therefore, a well-conceived study to provide the platform for optimal vaccine delivery will go a long way in providing the much-needed protection for the people and make the best use of the scarce resources available for the treatment of diseases. However, despite the well-conceived study, there are fundamental issues that should be taken into consideration and this will bother on i) recruitment of participants as well as ii) the long duration of follow-up. In this regard, care must be taken to ensure that the field workers involved in this study are carefully monitored to see that they adhere to all ethical considerations since those targeted are minors that may not necessarily know their rights and privileges concerning a study of this nature. Importantly, due to the long duration and follow-up, the investigators must take into consideration the contemporary challenges of exposing participants to undue risks bordering on COVID-19 and other associated factors in the environments where the study will be conducted. Overall, this is a well-conceived study that will only achieve the set goals if all the stakeholders meet up with their roles and the results are truly reported.
---

REVIEWER	Dr Lucy Ochola Institute of Primate Research, Nairobi, Kenya
REVIEW RETURNED	04-Aug-2020

GENERAL COMMENTS	dosage of praziquantel remains unclear in this protocol, and will diagnostic approaches be used to confirm parasite clearance? the same applies for malaria.
--

REVIEWER	Steve Kanters School of Population and Public Health, University of British Columbia, Canada
REVIEW RETURNED	31-Aug-2020

GENERAL COMMENTS	The protocol is well written and clear. The planned analyses tackle important scientific needs and the statistical methods are appropriate for them. I only had minor comments, as follows:  • On page 7:  o The language used in the Cohorts section is slightly confusing. Are we describing the cohorts from which the trial participants will be recruited or are we describing the participants in the trials as a cohort? The mixing of language for observational study design and RCT study design, it can leave the reader a bit confused. If this can be clarified, it would be helpful. The ask is not to avoid the language altogether. o Trial B is described as double-blind, placebo-controlled trial while the others (A and C) are not provided similar descriptions. Are we to gather that they are open-label? It would be helpful to the reader if additional clarity is provided • Something seems to have gone wrong with page numbering starting on page 10. Only the first digit appears. Could be a problem with how the PDF was saved? • On page 10, lines 247-264: There is some repetition of values regarding the significance level and expected loss to follow-up, might be useful to state these near the start and avoid the repetition • Page 10:  o The approaches to Objective iii could be clearer. What kinds of regressions are we expecting? I suspect logistic regression for BCG response and linear regression for the remaining outcomes, but are there some that are known a priori to require transformations (e.g. log transformations). o Will the different exposures (in utero, childhood, etc.) be analysed separately on together ? o Line 277: Infections erroneously has two ss at the end • Objective 4:  o It seems like response to BCG is the only outcome, but that isn't clear o An underlying assumption appears to be that persons are immutable with respect to setting. Will a person who was born in an urban setting but currently resides in a rural setting be included in the analysis or not? I imagine that the number of such patients is negligible at best, but am wondering if this was a concern that was thought of. • Supplementary, line 330: Do you mean Objective iv? If indeed objective 1, then use i instead so as to be consistent with the rest. Also capitalize Objective for further consistency
---

VERSION 1 – AUTHOR RESPONSE

COMMENTS FROM REVIEWER 1

Overall, a very well written and detailed study protocol on an important topic.

Some specific comments and suggestions:

Comment 1

Will you test for the actual presence of parasites before treatment and throughout the trials? Line 298 implies that this will be done, but I would recommend including this information (and the parasitological methods you plan to use) explicitly in the Methods section. If you don't already intend to test this, I would highly recommend that you consider it!

Response

We intend to test the presence of parasites before treatment and throughout the trials. This information is detailed in companion manuscripts that describe each of the three trials separately (bmjopen-2020-040426, bmjopen-2020-040427, and bmjopen-2020-040430), and which are now cited in this manuscript. We had not provided a detailed account of this information in the current manuscript (which describes the scientific rationale and cross-cutting analyses for all three trials); however, following the reviewer's comment, we have included this information in the Supplementary information for this manuscript.

Changes in manuscript: Line 130-139 (Supplementary information)

Comment 2

Line 100 – It seems to me that “trans-kingdom interaction” is a synonym for the more-widely used term “co-infection”, or at best it describes interactions that occur during co-infections. The only difference I can think of is that co-infection usually refers to co-habitation and interaction between two infectious organisms (parasites and pathogens), where trans-kingdom interactions seems to refer to interactions between pathogens and components of the microbiome. However, there isn't a clear line between microbiome and pathogens/parasites (and the 2016 Pfeiffer and Virgin “trans-kingdom interaction” paper cited here includes protozoa and helminths in the “microbiome” category), so I still think the terms are essentially synonymous. With this in mind, the literature on co-infections is far richer than the “trans-kingdom interaction” studies cited here, and I would recommend acknowledging its existence and citing some of these papers. I also recommend that you use the term “co-infection” alongside “trans-kingdom interactions” in your paper so that interested parties can find and read this paper.

Response

We interpret the ‘trans-kingdom’ concept as emphasising that the human (or mammalian) host actually hosts an ecosystem, which includes multicellular animals, such as worms, as well as protozoa, bacteria and viruses. These include both pathogens and commensal agents. The outcome for the host may be the result of trans-kingdom interactions between these, not just the direct effect of a single agent.

We have edited line 103-105 in the introduction to read as follows: The “trans-kingdom” concept emphasises that mammals support a complex ecosystem of multicellular organisms, such as helminths, as well as bacteria, fungi, protozoa and viruses, and suggests that these interact in their effects on the mammalian immune system, rather than acting alone, as individual agents.

Changes in the manuscript: Line 103-105

Comment 3

Line 120 – this is very certain phrasing and I would suggest couching it in some uncertainty since (I think?) it's your hypothesis, and not something that's been otherwise proven.

Response

We agree with the reviewer and have edited the sentence accordingly.

Changes in the manuscript: Line 123

Comment 4

Line 178 – typo: anthelminthic should be anthelmintic. When I searched, this spelling was used 4 times in the manuscript.

Response

Both spellings are commonly used, including by the World Health Organisation (<https://extranet.who.int/rhl/topics/preconception-pregnancy-childbirth-and-postpartum-care/antenatal-care/who-recommendation-preventive-anthelminthic-treatment>). We have also tended to use the spelling “anthelminthic” in our previous papers (such as <https://academic.oup.com/cid/article/68/10/1665/5093172>, <https://trialsjournal.biomedcentral.com/articles/10.1186/s13063-015-0702-5>, <https://journals.plos.org/plosone/article?id=10.1371/journal.pone.0050325>, <https://onlinelibrary.wiley.com/doi/full/10.1111/j.1399-3038.2010.01122.x>).

Changes in the manuscript: n/a

Comment 5

Line 199 – this is excellent, we sorely need more studies investigating the effects of parasites on different types of vaccine and on vaccines with different routes of administration. (In my recent meta-analysis on parasite-vaccine interactions I was unable to test for any patterns due to route of administration because of a lack of studies that used non-parenterally administered vaccines.)

Response

We appreciate the reviewer’s comment, and are looking forward to contributing to the literature on effects of parasites on different types of vaccines.

Changes in the manuscript: n/a

Comment 6

Line 276 – typo (“infectionss”)

Response

We thank the reviewer for noticing the typo. It has been rectified.

Changes in manuscript: Line 282

Comment 7

Line 306 – sentence ends abruptly – should this say "confounding factors"?

Response

We have edited line 319 in the main text accordingly.

Changes in manuscript: Line 319.

COMMENTS FROM REVIEWER 2

Comment 1

This work is properly conceived and well written. It took into consideration several population studies with diverse inferences across the globe, and more fundamentally, explores the nuances therein to set the stage for this work. As stated by the authors, "understanding the immunological predictors of vaccine response, and factors that drive them, will contribute to strategies for improving vaccine efficacy for rural, tropical settings". This can not be overemphasized judging from the fact that there are a plethora of infectious diseases afflicting people in sub-Saharan African where issues related to weak health infrastructure further undermines the ability to solve this huge public health debacle. Going forward, therefore, a well-conceived study to provide the platform for optimal vaccine delivery will go a long way in providing the much-needed protection for the people and make the best use of the scarce resources available for the treatment of diseases.

However, despite the well-conceived study, there are fundamental issues that should be taken into consideration and this will bother on i) recruitment of participants as well as ii) the long duration of follow-up. In this regard, care must be taken to ensure that the field workers involved in this study are carefully monitored to see that they adhere to all ethical considerations since those targeted are minors that may not necessarily know their rights and privileges concerning a study of this nature. Importantly, due to the long duration and follow-up, the investigators must take into consideration the contemporary challenges of exposing participants to undue risks bordering on COVID-19 and other associated factors in the environments where the study will be conducted.

Overall, this is a well-conceived study that will only achieve the set goals if all the stakeholders meet up with their roles and the results are truly reported.

Response

We thank the reviewer for their assessment of our work, and agree that a number of factors should be taken into consideration when conducting the study. The study team includes an internal monitor who ensures that all ethical considerations are adhered to and that all staff (including field workers) are sufficiently trained and monitored. Study site-initiation visits by the monitor have been conducted, or are planned (for work that has not commenced). We also hold staff training sessions prior to each trial timepoint, to refresh staff knowledge on areas such as ethics, study procedures and handling of adverse events. The study monitor, the PI, Project Leader and co-investigators work closely with the Research Compliance office of the MRC/UVRI and LSHTM Uganda Research Unit, which also conducts periodic audits of the study. Furthermore, we have invited an external monitor to conduct periodic, independent assessment of our activities.

We have carefully considered the challenges of exposing participants and staff to risks of COVID-19, and put measures in place to mitigate these risks. In addition, we have written a risk management plan (concerning conduct of our trial during the COVID-19 pandemic) that has been assessed and approved by the relevant local ethics committees, the Uganda National Drug Authority and the Uganda National Council for Science and Technology. Furthermore, all study staff have been trained on this plan.

Changes in manuscript: n/a

COMMENTS FROM REVIEWER 3

Comment 1

Dosage of praziquantel remains unclear in this protocol, and will diagnostic approaches be used to

confirm parasite clearance? the same applies for malaria.

Response

We have not provided detailed information on dosage of praziquantel in the current protocol manuscript (which describes the scientific rationale and cross-cutting analyses for all three trials). As mentioned in line 198-200, this information, akin to information on recruitment criteria, other interventions, randomisation and treatment allocation procedures, is detailed in protocols for Trials A, B and C (bmjopen-2020-040426, bmjopen-2020-040427 and bmjopen-2020-040430, respectively; published separately in this journal). Figure 3 and Figure 4 in the current protocol also summarise the trial interventions.

We intend to test the presence of parasites before treatment and throughout the trials. Tests will also confirm whether the parasites have been cleared, or not. Information on parasitological examinations is detailed in companion manuscripts that describe each of the three trials separately (bmjopen-2020-040426, bmjopen-2020-040427, and bmjopen-2020-040430), and which are now cited in this manuscript. We had not provided a detailed account of this information in the current manuscript (which describes the scientific rationale and cross-cutting analyses for all three trials); however, following the reviewer's comment, we have included this information in the Supplementary information for this manuscript.

Changes in manuscript: Line 130-139 (Supplementary information)

COMMENTS FROM REVIEWER 4

The protocol is well written and clear. The planned analyses tackle important scientific needs and the statistical methods are appropriate for them. I only had minor comments, as follows:

Comment 1

On page 7: The language used in the Cohorts section is slightly confusing. Are we describing the cohorts from which the trial participants will be recruited or are we describing the participants in the trials as a cohort? The mixing of language for observational study design and RCT study design, it can leave the reader a bit confused. If this can be clarified, it would be helpful. The ask is not to avoid the language altogether.

Response

Text in the 'Cohorts' section has been edited to clarify the description of participants and the study settings (Trial A and B) and cohort (Trial C) from which they will be recruited.

Changes in manuscript: Line 175-185

Comment 2

Trial B is described as double-blind, placebo-controlled trial while the others (A and C) are not provided similar descriptions. Are we to gather that they are open-label? It would be helpful to the reader if additional clarity is provided

Response

Additional clarity has been provided.

Changes in manuscript: Line 187-193

Comment 3

Something seems to have gone wrong with page numbering starting on page 10. Only the first digit appears. Could be a problem with how the PDF was saved?

On page 10, lines 247-264: There is some repetition of values regarding the significance level and expected loss to follow-up, might be useful to state these near the start and avoid the repetition

Response

We thank the reviewer for noticing the repetitions. The text has been edited accordingly.

Changes in manuscript: Line 267-269

Comment 4

Page 10: The approaches to Objective iii could be clearer. What kinds of regressions are we expecting? I suspect logistic regression for BCG response and linear regression for the remaining outcomes, but are there some that are known a priori to require transformations (e.g. log transformations).

Response

We have included more information in the section on the Approach to Objective iii, to make it clearer.

Changes in manuscript: Line 282-291

Comment 5

Page 10: Will the different exposures (in utero, childhood, etc.) be analysed separately or together?

Response

We will use a hierarchical statistical modelling approach so that early life exposures are not adjusted for later life exposures that may be on the causal pathway to the outcome, but associations between later life exposures and vaccine response outcomes will be adjusted for early life exposures to control for these as potential confounding factors. We have included this information in the main text (Line 283-286)

Changes in manuscript: Line 283-286

Comment 6

Line 277: Infections erroneously has two ss at the end

Response

We thank the reviewer for noticing the typo. It has been rectified.

Changes in manuscript: Line 282

Comment 7

Objective 4: It seems like response to BCG is the only outcome, but that isn't clear

Response

As mentioned in the abstract, and in line 207-211, we will study a portfolio of licensed vaccines expected to be beneficial (some already given) to adolescents in Uganda: live parenteral (BCG, yellow fever), live oral (typhoid) and non-live (HPV [a viral particle vaccine] and tetanus/diphtheria [Td; toxoid vaccines]). This will allow us to compare study outcomes (responses to several vaccines) between three Ugandan settings (using data from all Trials): rural, high schistosomiasis exposure; rural, high malaria exposure; and urban (Objective 4).

Changes in manuscript: n/a

Comment 8

Objective 4: An underlying assumption appears to be that persons are immutable with respect to setting. Will a person who was born in an urban setting but currently resides in a rural setting be included in the analysis or not? I imagine that the number of such patients is negligible at best, but am wondering if this was a concern that was thought of.

Response

The reviewer raises a concern that we have thought about. Therefore, our participant questionnaires will capture information on residence, including where individuals were born, where they lived in the first five years of life and, since our enrolment is school-based, where they spend their school holidays. If there are a large number of participants whose residence has changed (which we do not expect) then we will conduct sensitivity analyses to investigate the impact of this.

Changes in the manuscript: n/a

Comment 9

Supplementary, line 330: Do you mean Objective iv? If indeed objective 1, then use i instead so as to be consistent with the rest. Also capitalize Objective for further consistency

Response

Many thanks to the reviewer for pointing this out. Indeed, we mean Objective iv, and have corrected this. The word Objective has been capitalized, for consistency.

Changes in the manuscript: Lines 348, 358 (Supplementary information)

VERSION 2 – REVIEW

REVIEWER	Liana Wait Princeton University, United States
REVIEW RETURNED	03-Nov-2020

GENERAL COMMENTS	The author appears to have addressed my (and the other reviewers') concerns.
--

REVIEWER	Steve Kanters University of British Columbia
REVIEW RETURNED	05-Oct-2020

GENERAL COMMENTS	The protocol is well written and appropriate for publication.
---